# The Prevalence of Anticitrullinated Protein Antibodies in Older Poles—Results from a Population-Based PolSenior Study

**DOI:** 10.3390/ijerph192114216

**Published:** 2022-10-31

**Authors:** Anna Chudek, Przemysław Kotyla, Małgorzata Mossakowska, Tomasz Grodzicki, Tomasz Zdrojewski, Magdalena Olszanecka-Glinianowicz, Jerzy Chudek, Aleksander J. Owczarek

**Affiliations:** 1Health Promotion and Obesity Management Unit, Department of Pathophysiology, Faculty of Medical Sciences in Katowice, Medical University of Silesia in Katowice, 40-752 Katowice, Poland; 2Department of Internal Medicine, Rheumatology and Clinical Immunology, Faculty of Medical Sciences in Katowice, Medical University of Silesia in Katowice, 40-635 Katowice, Poland; 3Study on Ageing and Longevity, International Institute of Molecular and Cell Biology, 02-109 Warsaw, Poland; 4Department of Internal Medicine and Gerontology, Jagiellonian University Medical College, 31-531 Krakow, Poland; 5Division of Preventive Medicine and Education, Medical University of Gdansk, 80-211 Gdansk, Poland; 6Department of Internal Medicine and Oncological Chemotherapy, Medical University of Silesia, 40-029 Katowice, Poland

**Keywords:** older adults, anti-CPP prevalence, rheumatoid arthritis, ACPA, anticitrullinated protein antibodies

## Abstract

Little is known about the occurrence of antibodies in older subjects. We analyzed the prevalence of anticitrullinated protein antibodies (anti-CCP) in a representative cohort of Polish older adults, participants of PolSenior substudy. Randomly selected 1537 serum samples of community-dwelling participants aged 65 and over. Questionnaires were completed by qualified interviewers and laboratory assessments served as a database for this analysis. The frequency of anti-CCP seropositivity (N = 50) was estimated at 3.25% (95% CI: 2.45–4.30%), being higher among women—4.05% (2.83–5.73%) than men—2.41% (1.48–3.86%). The frequency of anti-CCP seropositivity was decreasing with age from 4.29% in aged 65–74 years and 4.07% in 70–84 years to 1.50% in aged 85 years or above (*p* < 0.05). Hypoalbuminemia, inflammatory status (C-reactive protein >10 mg/dL or interleukin-6 ≥10 pg/mL), and female gender were associated with increased, while age ≥85 years with decreased risk of seropositivity. Multivariable logistic regression revealed that hypoalbuminemia, inflammatory status, and age ≥85 years were independently associated factors of anti-CCP seropositivity. The decreased frequency of anti-CCP seropositivity in the oldest old suggests shorter survival of the seropositive individuals who developed rheumatoid arthritis. It seems that low symptomatic RA remains frequently undiagnosed in older subjects.

## 1. Introduction

Anticitrullinated protein antibodies (ACPA, tested usually as anti-cyclic citrullinated peptide—anti-CCP) seropositivity is a common feature of rheumatoid arthritis (RA) and a part of the joined American College of Rheumatology and European League Against Rheumatism classification criteria for this disease [1]. They are detected in 55–69% of symptomatic patients with RA [2,3]. Of note, the occurrence of anti-CCP antibodies is more specific than of rheumatoid factor (RF) for the RA diagnosis. The specificity of anti-CCP antibodies is estimated at 90–98%, while the specificity of the RF is at 68–85% [4,5]. In addition, the occurrence of anti-CCP, similarly to RF, may precede the clinical manifestation of RA even by more than 10 years [6,7]. It was also shown that the occurrence of ACPA in patients with RA, for the longer course of the disease, was associated with a more severe clinical course, increased frequency of symptoms related to swelling of the joints and their radiological damage (bone erosions), and poorer prognosis [8,9].

Anti-CCP antibodies cannot be considered as pathognomonic for RA, as they are also detected in other disease entities such as palindromic rheumatism, psoriatic arthritis, systemic lupus erythematosus, systemic sclerosis, Sjogren’s syndrome, polymyositis, dermatomyositis, ulcerative colitis, HCV seropositive patients with arthralgia and also in some infectious diseases [10,11,12,13,14,15,16,17,18,19,20,21,22]. Furthermore, anti-CCP antibodies are detected in apparently healthy individuals and blood donors with a frequency of about 1% [23].

Little is known about the presence of anti-CCP antibodies in the general population. In a population-based study in Japan (N = 9575), the prevalence of anti-CPP positivity increased from 1.3% in middle age to 3.0% in aged 70–75 yrs [24]. The Turkish study (N = 941) with a low percentage of older subjects (≥60 yrs: 14.3%) showed a prevalence of 1.0% [25].

In Europe, there were 1.0% seropositive middle-aged participants in the Netherlands study (N = 40,136) [26], and 2.8% in the Swedish twin study (N = 12,590) middle age and older cohort including patients with rheumatoid arthritis [27].

Only a few associates of anti-CPP positivity were identified: older age, female gender, smoking, joint complaints, including rheumatoid arthritis [24,26].

To the best of our knowledge, there is no data concerning the prevalence of anti-CCP antibodies in Poland and other Central European countries.

Therefore, the current investigation aims to analyze the prevalence and associates of anti-CPP antibodies occurrence in the older Polish population.

## 2. Materials and Methods

### 2.1. Study Population

The analysis was a substudy of a large nationwide, multicenter, interdisciplinary project on ageing in Poland (PolSenior1), performed between 2008 and 2010. For the study protocol details see Bledowski et al. [28]. In brief, the project recruited 4979 subjects in 7 age cohorts (65–69 yrs., 70–74 yrs., 75–79 yrs., 80–84 yrs., 85–89 yrs., and 90 yrs. or over), a group representative for Polish older population. The protocol was approved by the Bioethics Committee of the Medical University of Silesia (KNW-6501-38/I/08). All subjects or their caregivers signed informed consent before the study assessments were carried out.

As shown on the analysis flow chart (Figure 1), the anti-CCP antibodies titer was assessed in a randomly selected subgroup of 1537 samples (37.5%). The initial analysis compared assessed (N = 1537) and not assessed (N = 2560) subjects for the anti-CCP antibodies titer to prove representatives of the analysis’s subgroup. We have excluded subjects who refused blood sample donation.

### 2.2. Laboratory Assessments

All biochemical assessments were performed in frozen samples. Anti-CCP antibodies were tested at the Institute of Rheumatology (currently National Geriatrics, Rheumatology and Rehabilitation Institute). The serum titer of anti-CCP antibodies was measured by using the Anti-CCP Immunoscan RA ELISA kit (Euro Diagnostica AB, Malmö, Sweden). Values ≥ 25 U/mL were scored as positive. Serum 25-hydroxycholecalciferol levels (25(OH)D) levels were measured by HYBRID XL analyzer (DRG Instruments GmbH, Marburg, Germany) with a limit of quantification (LoD) of 4.6 ng/mL and intraassay precision of 14.2% for lower 25(OH)D concentrations. Plasma interleukin-6 (IL-6) was measured by ELISA (R&D Systems, Minneapolis, MN, USA) with a LoD of 0.04 pg/mL and mean intraassay and intraassay coefficient of variance <7.8% and <7.2%, respectively.

Other laboratory assessments were performed in a selected for the PolSenior project laboratory in Warsaw. Serum total cholesterol, LDL and HDL fractions, triglycerides, glucose, albumin, creatinine, C-reactive protein (CRP), and uric acid concentrations were assessed by an automated system (Modular PPE, Roche Diagnostics GmbH, Mannheim, Germany) with intraassay coefficients of variability below 1.7%, 1.2%, 1.3%, 1.8%, 1.7%, 1.7%, 2.3%, 5.7%, and 1.7%, respectively. The presence of anti-HCV antibodies was determined by electrochemistry assay on Elecsys 2010 analyzer (Roche Diagnostics GmbH, Mannheim, Germany). Hemoglobin concentrations were measured in local routine laboratories with different analyzers. Hematuria was diagnosed using Combur-Test strip technology (Roche Diagnostics Miditron M ChemStrip Urine Analyzer). The albumin/creatinine ratio (ACR) and estimated glomerular filtration rate (eGFR) were calculated based on the enquired results of the above laboratory tests.

### 2.3. Data Analysis

Nutritional status, according to WHO criteria, was defined based on body mass index (BMI) as: underweight (<18.5 kg/m^2^), normal weight (18.5 ÷ 24.9 kg/m^2^), overweight (25.0 ÷ 29.9 kg/m^2^), and obesity (≥30.0 kg/m^2^) [29]. The waist circumference (WC) thresholds for abdominal obesity in the Caucasian population were according to the definition of the International Diabetes Federation—IDF (≥94 cm in men and ≥80 cm in women) [30].

The risk of malnutrition was assessed with the Mini Nutritional Assessment-Short Form (MNA-SF) and classified as: malnutrition (≤7 pts), risk of malnutrition (8 ÷ 11 pts), and normal nutritional status (≥12 pts) [31].

Hypertension was diagnosed based on an average from four measurements performed on two separate visits, according to the 2013 ESH/ESC Guidelines for the Management of Arterial Hypertension [32], values equal to or higher than 140 mmHg (systolic BP) and/or 90 mmHg (diastolic BP), or antihypertensive treatment, as described previously [33].

Diabetes was defined as a fasting plasma glucose level ≥126 mg/dL or the use of antidiabetic medications [34]. Congestive heart failure (CHF), coronary artery disease (CAD), chronic obstructive pulmonary disease (COPD), and asthma were scored based on the participants’ reports and hospital discharge cards. Selected groups of medications (non-steroidal anti-inflammatory drugs—NSAIDs, aspirin, glucocorticoids, disease-modifying anti-rheumatic drugs—DMARDs), based on the participants’ reports, were included in the analysis.

Hypercholesterolemia was defined as total cholesterol greater than 190 mg/dL or the use of statins and hypertriglyceridemia as serum triglyceride (TG) level above 150 mg/dL or the use of fibrates [35].

Hyperuricemia was defined as serum uric acid level above 6 mg/dL in women and 6.8 mg/dL in men (population sex-specific ranges corrected in men for the uric acid solubility in water), or the use of allopurinol—the only used xanthine oxidase inhibitors in the PolSenior1 cohort [36].

Chronic pain was defined as pain that lasted more than 3 months [37].

The Lawton Instrumental Activities of Daily Living Scale (IADL) was used to determine the functional status of participants. The score of 8–18 points was classified as dependent, 19–23 points as partially dependent, and 24 points as independent [38].

Moderate-to-severe anemia was defined by the concentration of blood hemoglobin below 11 g/dL [39].

As hypoalbuminemia, we assumed the level of albumins below the lower limit of the normal range (35 g/L).

The C-reactive protein level above 3 mg/dL was considered as a marker of high cardiovascular risk and CRP level above 10 mg/dL indicated inflammatory status [40]. Interleukin 6 cut-off value was established as a value for 95% of our study cohort.

Albumin-to-creatinine ratio was used to estimate the occurrence of albuminuria and the cut-off point was >30 mg/g. The eGFR was estimated according to the full MDRD formula [41]. Two eGFR ranges were used: below 60 mL/min/1.73 m^2^ corresponding to G3a–G5 and below 45 mL/min/1.73 m^2^ corresponding to stage G3b–G5 of chronic kidney disease (CKD) according to NKF KDOQI guidelines [42].

### 2.4. Disability

#### 2.4.1. Sociodemographic Variables

The demographic and socioeconomic status was assessed using data derived from the questionnaire (place of residence, living conditions, type of work in the past, personal income, frequency of alcohol consumption, smoking status with packyears of exposure). Net personal income was voluntarily reported and referred to the average retirement pension of 1000 PLN in Poland in 2009, as low (below 100% of average), moderate (over but below 200% of average), and high (over 200% of average). Participants reporting alcohol consumption ≥3 times per month were scored as ‘alcohol consumers’.

#### 2.4.2. Statistical Analysis

Statistical Analysis was performed using STATISTICA 13.0 PL (TIBCO Software Inc., Palo Alto, CA, USA), StataSE 13.0 (StataCorp LP, College Station, TX, USA), and R software [R Core Team (2013). R: A language and environment for statistical computing. R Foundation for Statistical Computing, Vienna, Austria. URL http://www.R-project.org/; accessed on 1 September 2020]. Statistical significance was set at a *p*-value below 0.05. All tests were two-tailed. Imputations were not done for missing data. Nominal and ordinal data were expressed as percentages. Interval data were expressed as mean value ± standard deviation in the case of normal distribution. In the case of data with skewed or non-normal distribution, it was expressed as median, with lower and upper quartiles. The distribution of variables was evaluated by the Anderson-Darling test and the quantile-quantile (Q-Q) plot. The homogeneity of variances was assessed by the Levene test. Comparisons between two groups were done with the Student t-test for independent groups in case of interval data. Nominal and ordinal data were compared with the χ^2^ test. Risk factors of the anti-CCP positive result were evaluated with univariable logistic regression. Then, the best model of multivariable logistic regression was done. Results were presented as corresponding odds ratios (OR) with confidence intervals (±95% CI) and *p*-values.

## 3. Results

### 3.1. Characteristics of the Sub-Study Population

The analyzed substudy cohort included 791 (51.5%) women and 746 (48.5%) men, one-third aged 85 years or more—Table 1. Despite random sample selection, we did not avoid some, although small, differences. In comparison to the subgroup without assessed anti-CCP antibodies, this subgroup includes more women and was two years older due to an overrepresentation of aged 85 years or above. There were more rural residents living alone, and ‘white collars’ but fewer subjects with low personal incomes. More participants in this subgroup had malnutrition, vitamin D deficiency, and a history of hospitalization for CHF. On the contrary, they were less suffering from chronic lung disease (COPD or asthma) and had hypercholesterolemia and hyperuricemia.

There was no difference in the utilization of aspirin, non-steroidal anti-inflammatory drugs (NSAIDs), glucocorticoids (GKS), and disease-modifying antirheumatic drugs (DMARDs) between assessed and not-assessed groups. Moreover, both substudy cohorts had similar percentages of laboratory tests declared to be done within the last 3 years and hospitalization for any cause during the last 5 years. Finally, the 5-year overall survival rates were similar.

### 3.2. Prevalence of Anticitrullinated Protein Antibodies

There were 50 anti-CCP seropositive participants. The mean level of anti-CCP antibodies was 28.1 ± 15.2 U/mL (range: 3–55). The frequency of anti-CCP seropositivity was estimated at 3.25% (95% CI: 2.45–4.30%), being higher among women—4.05% (95% CI: 2.83–5.73%) than men—2.41% (95% CI: 1.48–3.86%). The frequency of anti-CCP seropositivity was decreasing with age from 4.29% in aged 65–74 years and 4.07% in aged 70–84 years to 1.5% aged 85 years or above (*p* < 0.05).

### 3.3. Characteristics of Anti-CCP Seropositive Subjects

There were more women among seropositive subjects, especially in the oldest subgroup (Figure 2), yet the difference did not reach statistical significance (*p* = 0.27). A higher percentage of women in the anti-CCP positive subgroup explains a lower percentage of alcohol consumers among these subjects (Table 1). Seropositive subjects were younger by 4 years, with a lower percentage of subjects aged 85 years or above. They were characterized by higher levels of CRP and IL-6, and lower levels of albumin. Increased CRP > 10 mg/dL or IL-6 > 10 pg/mL were observed in 16.3% and 20.8%, respectively. The frequency of hypoalbuminemia was more than 5-time higher in seropositive subjects (15.2%). There were no differences in the occurrence of hypertension, CHF, CAD, COPD, or asthma between seropositive and seronegative subjects.

There were 36 subjects with markedly increased CRP and IL-6 levels in the whole study group. Four (9.5%) were in the anti-CCP seropositive subgroup and 32 (2.3%) in the anti-CCP seronegative subgroup (*p* < 0.05).

Seropositive subjects with markedly increased CRP and/or IL-6 levels more frequently reported chronic pain than seronegative ones (60.0% vs. 40.6%). The most frequent locations of pain in seropositive subjects were the lumbar region (28.0%), knees (22.0%), and feet (12.0%).

Four seropositive subjects were treated with steroids and/or DMARDs (GKS monotherapy—N = 2; DMARDs monotherapy—N = 1; GKS + DMARDs—N = 1). Half of them had increased CRP or IL-6 levels.

### 3.4. Associates of Anti-CCP Seropositivity

Factors related to the risk of anti-CCP seropositivity were defined based on the logistic regression (Table 2). The risk increased with hypoalbuminemia, higher serum CRP level (as well as CPR > 3 mg/dL) or plasma IL-6 level (as well as IL-6 ≥ 10 pg/mL), inflammatory status (CRP > 10 mg/dL or IL-6 ≥ 10 pg/mL), and female gender. While older age (as well as age ≥85 yrs), serum albumin levels and alcohol consumption decreased the risk of seropositivity.

Multivariable logistic regression revealed that hypoalbuminemia, inflammatory status, and age ≥85 yrs were independently associated factors of seropositivity (Table 2).

#### Survival Analysis

No significant difference in 5-year survival rates was noted between seropositive and seronegative participants. The parallel Kaplan-Meier survival curve analysis disclosed no difference (p_log-rank_ = 0.85).

## 4. Discussion

This analysis included a part of the population-based cohort of Polish older adults, participants of the PolSenior project, with a relatively large group of octogenarians and nonagenarians. The main aim of our study was to analyze the prevalence of anti-CCP seropositivity and its correlates in the elderly population. We estimated the frequency of ani-CCP seropositivity at 3.25% in Polish older adults, which is comparable to the value obtained in the 70–75 yrs old Japanese participants of the Terao study—3.0% [24]. As expected, the frequency of seropositivity in our cohort was greater among women (4.05%), than men (2.41%), similarly to other studies [24,26].

The majority of previous studies showed that the presence of anti-CCP antibodies in the population is increasing with age from about 1.0–1.3% in middle-aged people [24,26] to approximately 3.0% in aged 70–75 yrs [24]. In our study, the frequency of seropositivity in 65–74 yrs and 70–84 yrs old subgroups was even higher (4.3% and 4.1%, respectively) with a marked decrease in the oldest—85 yrs and older (1.5%).

To the best of our knowledge, there is no study assessing the prevalence of anti-CCP seropositivity in a very old population. Of note, JunSoo Ro et al. described the decreasing trend in the frequency of RA in octogenarians based on data retrieved from the Korean National Health Insurance Service (NHIS) [43]. The declining prevalence of seropositivity in the oldest people shown in our study may be caused by accelerated atherosclerosis and a higher prevalence of osteoporosis, oncological, cardiovascular, and pulmonary diseases, as well as neurodegenerative disorders in individuals suffering from RA [44,45], leading to excessive premature mortality. However, our 5-year survival analysis failed to support the shorter survival of seropositive subjects, which may be caused by a small subset of anti-CPP positive, declining the power of statistical analysis. Due to the small sample size, we did not analyze the influence of age and the presence of RA symptoms in the survival analysis. Moreover, we noticed a similar prevalence of hypertension, CAD, CHF, and chronic pulmonary diseases in the seropositive and seronegative subgroups.

Smoking is a well-established risk factor predisposing to the development of RA, first of all by triggering immune reactions to autoantigens modified by citrullination in genetically pre-disposed subjects, and also by stimulation of inflammation and oxidative stress [46]. In our study, the percentage of active smokers was slightly but not significantly higher in anti-CCP positive subjects, possibly due to the small size of the study group.

Our study triggers the discussion concerning undiagnosed low-symptomatic RA in the population of older adults. Bone and joint complaints, usually attributed to osteoarthritis in seniors, frequently do not result in performing diagnostic workups for inflammatory arthritis. That may result in some undiagnosed cases. In our project, the lack of RF titer, joint imaging, and clinical evaluation of joints in a physical examination precludes the application of the European Alliance of Associations for Rheumatology (EULAR) or the American College of Rheumatology (ACR) diagnostic criteria for RA. Therefore, we can only estimate the frequency of anti-CCP positive subjects with inflammation (CRP > 10 mg/dL and/or IL-6 ≥ 10 pg/mL) and point out the increased risk of seropositivity in subjects with low serum albumin levels. Such inflammatory profile presented one-fifth of 50 anti-CCP seropositive subjects. Of those, five (10%) were treated with GKS and/or DMARDs and 46% declared chronic pain. This data suggest that a large group of older adults remains undiagnosed with RA, and thus not properly treated. It may be caused by atypical joint symptoms, with more prevalent knee and shoulder involvement in patients with late-onset RA (who developed RA at age over 65 years) [47].

We should have in mind that anti-CCP antibodies can be detected in numerous disease entities other than RA, and also in apparently healthy people, which points to limitations in specificity in RA diagnosis. Therefore, we cannot precisely estimate the scale of the problem of undiagnosed RA, which is also poorly recognized in the literature. Further studies are needed on this topic with the participation of qualified rheumatologists and the assessment of all diagnostic criteria included in the current guidelines. In addition, asymptomatic anti-CCP seropositive patients may benefit by being followed up more closely. Moreover, it is suggested that the occurrence of periodontitis may predispose to the formation of anti-CCP antibodies [48]. However, the periodontal status was not assessed, we cannot exclude that decreasing seropositivity is not related to the high rate of edentulism in the oldest [49].

The main limitations of our study were mentioned above, however, it was designed to cover the most important aspects of ageing, yet not included, all signs and symptoms required to diagnose RA. Nevertheless, the measurement of anti-CCP antibodies titer in a large community-based cohort with a high representation of the oldest-old enabled us to show a decreased frequency of seropositivity and its risk factors in such a unique population.

## 5. Conclusions

The decreased frequency of anti-CCP seropositivity in the oldest old suggests shorter survival of the seropositive individuals who developed rheumatoid arthritis. It seems that low symptomatic RA remains frequently undiagnosed in older subjects.

## Figures and Tables

**Figure 1 ijerph-19-14216-f001:**
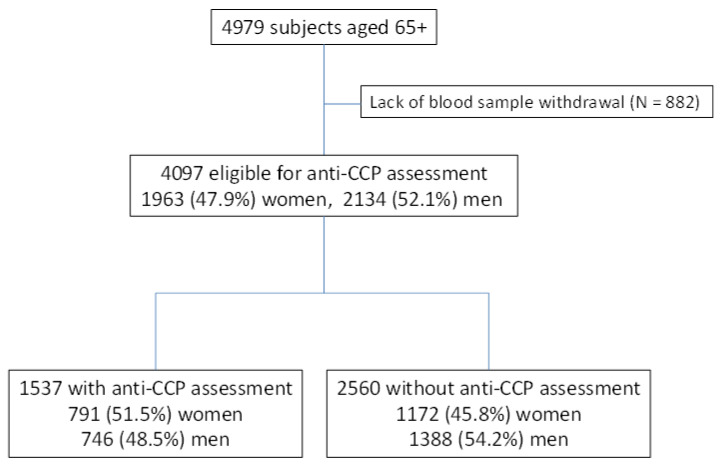
Flow chart.

**Figure 2 ijerph-19-14216-f002:**
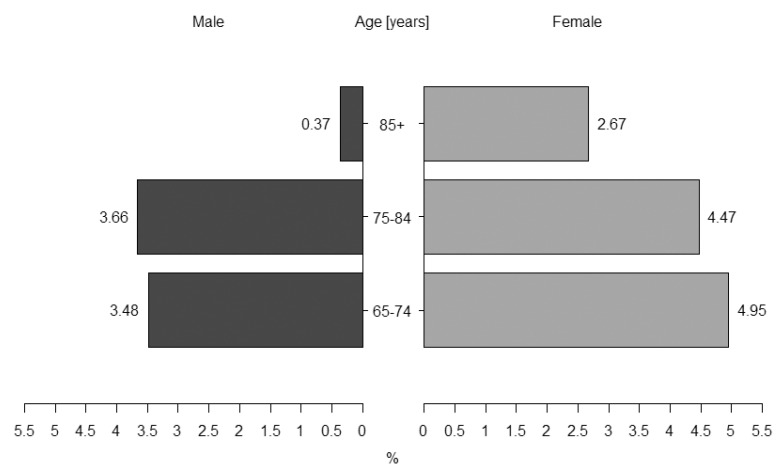
Percentage of anti-CCP positive males and females in age-related subgroups.

**Table 1 ijerph-19-14216-t001:** Characteristics of study subjects with not assessed and assessed anti-CCP antibodies (including positive and negative subgroups) that were enclosed in the analysis.

	Subgroup with Not Assessed Anti-CCP [N = 2560] 62.5%	Subgroup with Assessed Anti-CCP [N = 1537] 37.5%	Anti-CCP Negative [N = 1487] 96.75%	Anti-CCP Positive [N = 50] 3.25%
Women [N; %]	1172; 45.8	791; 51.5 ^#^	759; 51.0	32; 64.0
Age [years]	78 ± 9	80 ± 9 ^#^	80 ± 9	76 ± 9 *
65–74 years [N; %]	969; 37.8	513; 33.4 **	491; 33.0	22; 44.0
75–84 years [N; %]	858; 33.5	492; 32.0	472; 31.7	20; 40.0
≥85 years [N; %]	733; 28.6	532; 34.6 ^#^	524; 35.2	8; 16.0 **
Rural area residence [N; %]	1447; 56.5	999; 65.0 ^#^	969; 65.2	30; 60.0
Living alone [N; %]	487; 19.4	336; 22.4 *	326; 22.4	10; 20.8
Blue-collar [N; %]	1326; 56.8	788; 56.8	762; 56.8	26; 57.8
White-collar [N; %]	651; 27.9	431; 31.1 *	417; 31.1	14; 31.1
Personal income				
Low [N; %]	943; 42.2	501; 36.9 **	485; 36.9	16; 38.1
Average [N; %]	1083; 48.5	708; 52.2 *	685; 52.1	23; 54.8
High [N; %]	207; 9.3	147; 10.8	144; 11.0	3; 7.1
Alcohol consumers [N; %]	1619; 64.3	966; 64.0	941; 64.4	25; 50.0 *
Women	557; 48.6	399; 51.5	387; 52.1	12; 37.5
Men	1062; 77.5	567; 77.1	554; 77.3	13; 72.2
Active smokers [N; %]	257; 10.0	129; 8.4	123; 8.3	6; 12.0
Past smokers [N; %]	885; 34.6	533; 34.7	517; 34.8	16; 32.0
Packyears	31 (14–48)	28 (12–45)	28 (12–45)	31 (24–44)
BMI [kg/m^2^]	28.2 ± 5.0	28.1 ± 5.1	28.0 ± 5.1	29.3 ± 5.6
Overweight [N; %]	1012; 41.5	566; 39.3	547; 39.3	19; 41.3
Obesity [N; %]	793; 32.6	456; 31.7	438; 34.4	18; 39.1
Visceral obesity [N; %]	2027; 82.0	1179; 79.7	1136; 79.4	43; 87.8
Risk of malnutrition [N; %]	982; 41.4	599; 42.8	581; 42.9	18; 39.1
Malnutrition [N; %]	234; 9.9	175; 12.5 *	171; 12.6	4; 8.7
Dependent in IADL [N; %]	275; 10.8	193; 12.6	185; 12.5	8; 16.0
Chronic pain [N; %]	1108; 43.4	626; 41.1	603; 40.9	23; 46.0
Hospitalization in the past 5 yrs [N; %]	1363; 55.5	759; 51.7	738; 51.9	21; 43.8
Lab tests performed in the last 3 yrs [N; %]	1382; 55.6	820; 55.1	794; 55.1	26; 54.2
Comorbidity				
Diabetes [N; %]	577; 22.6	353; 23.6	340; 23.4	13; 27.7
Hypertension [N; %]	1889; 74.0	1093; 71.6	1060; 71.8	33; 66.0
Coronary artery disease [N; %]	568; 22.2	307; 20.0	297; 20.0	10; 20.0
Congestive heart failure [N; %]	134; 5.4	112; 7.3 *	109; 7.3	3; 6.1
COPD/Asthma [N; %]	490; 19.1	243; 15.8 **	234; 15.7	9; 18.0
Hypercholesterolemia [N; %]	1928; 75.3	1068; 71.0 **	1038; 71.3	30; 61.2
Hypertriglyceridemia [N; %]	677; 26.5	358; 24.1	343; 23.8	15; 32.6
Hyperuricemia [N; %]	755; 29.5	386; 26.0 *	373; 25.9	13; 28.3
Anti-HCV positive [N; %]	75; 3.1	32; 2.4	31; 2.4	1; 2.4
Hb [g/dL]	13.7 ± 1.5	13.8 ± 1.5 *	13.8 ± 1.5	14.1 ± 1.4
Hb < 11.0 g/dL [N; %]	81; 3.8	34; 2.7	34; 2.8	0
Albumin [g/L]	42.8 ± 3.4	42.2 ± 3.5 ^#^	42.2 ± 3.4	40.5 ± 5.1 *
Albumin < 35 g/L [N; %]	62; 2.4	46; 3.1	39; 2.7	7; 15.2 ^#^
eGFR [mL/min/L.73m^2^]	66.4 ± 17.8	64.8 ± 19.3 **	64.8 ± 19.3	65.1 ± 21.1
< 60 mL/min/L.73m^2^ [N; %]	893; 34.9	584; 39.3	565; 39.2	19; 41.3
< 45 mL/min/L.73m^2^ [N; %]	315; 12.3	220; 14.8	213; 14.8	7; 15.2
ACR > 30 mg/g [N; %]	384; 16.2	232; 16.5	228; 16.7	4; 9.1
Hematuria [N; %]	228; 10.1	122; 9.8	121; 10.0	1; 2.6
CRP [mg/dL]	2.36 (1.16–5.04)	2.32 (1.04–4.78)	2.30 (1.04–4.70)	3.90 ** (1.64–7.85)
> 3 mg/dL [N; %]	1064; 41.6	615; 41.7	589; 41.1	26; 60.5 *
> 10 mg/dL [N; %]	305; 11.9	154; 10.4	147; 10.3	7; 16.3
IL-6 [pg/mL]	2.3 (1.5–3.9)	2.4 (1.6–3.8)	2.3 (1.6–3.8)	2.9 ** (2.1–5.8)
≥ 10 pg/mL [N; %]	128; 5.4	82; 5.5	72; 5.0	10;20.8 ^#^
CRP or IL6 > 10 [N; %]	323. 14.0	182. 12.7	172; 12.4	10; 20.8 *
25-OH-D [ng/mL]	19.3 (14.4–25.1)	18.7 * (14.0–24.1)	18.6 (14.0–24.2)	19.4 (12.2–24.0)
25-0H-D < 20 ng/mL [N; %]	1108; 52.7	736; 57.2 **	713; 57.4	23; 51.1
Uric acid [mg/dL]	5.3 (4.5–6.5)	5.3 (4.3–6.4)	5.3 (4.4–6.4)	5.6 (4.5–6.3)
Selected medication				
NSAIDs [N; %]	370; 14.7	215; 14.2	205; 14.0	10; 20.0
Aspirin [N; %]	870; 34.6	505; 33.3	483; 32.9	22; 44.0
GKS [N; %]	29; 1.1	10; 0.7	7; 0.5	3; 6.0 ^#^
DMARDs [N; %]	9; 0.4	4; 0.3	2; 0.1	2; 4.0
5-year survival [N; %]	1694; 66.2	991; 64.5	954; 64.2	37; 74.0

DMARDs (disease-modifying antirheumatic drugs) = sulfasalazine or methotrexate; GKS (glucocorticoids). mean ± standard deviation or median (lower–upper quartile); * *p* < 0.05; ** *p* < 0.01; ^#^
*p* < 0.001; comparison of subgroups with and without assessed anti-CCP, positive and negative anti-CCP.

**Table 2 ijerph-19-14216-t002:** Results of univariable and multivariable logistic regression analysis for anti-CCP positivity.

	Univariable Analysis			Multivariable Analysis		
	OR	±95% CI	*p*	OR	±95% CI	*p*
Women	1.70	0.95–3.06	0.07	–		
Age per 10 years	0.65	0.47–0.91	<0.05	NI		
65–74 years	Ref			Ref		
75–84 years	0.94	0.51–1.76	0.86	0.67	0.34–1.34	0.26
≥85 years	0.34	0.15–0.77	<0.05	0.19	0.08–0.48	<0.001
Active smokers	1.47	0.60–3.64	0.40			
Alcohol consumers	0.55	0.31–0.97	<0.05	0.54	0.29–1.01	0.06
Albumin per g/L	0.89	0.83–0.96	<0.01	NI		
Albumin < 35 g/L	6.46	2.72–15.34	<0.001	6.29	2.33–16.98	<0.001
CRP per mg/dL	1.033	1.013–1.053	<0.01	NI		
>3 mg/dL	2.19	1.18–4.07	<0.05	NI		
>10 mg/dL	1.70	0.74–3.89	0.21	NI		
IL-6 per pg/mL	1.079	1.024–1.137	<0.01	NI		
≥10 pg/mL	4.98	2.39–10.40	<0.001	NI		
CRP or IL-6 ≥ 10	1.49	1.03–2.14	<0.05	6.14	2.70–14.00	<0.001

NI—not included.

## Data Availability

The data presented in this study are available on request from the corresponding author.

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
