# Peer review of "The Prevalence of Anticitrullinated Protein Antibodies in Older Poles—Results from a Population-Based PolSenior Study"

_ijerph, 2022, doi:10.3390/ijerph192114216_

Round 1

Reviewer 1 Report

The authors analyze the prevalence and associates of anti-CPP antibodies occurrence in the Polish older population. I think that it is interesting to assess the prevalence and associates of anti-CCP antibody occurrence in such a unique population. However, I think the authors need some work and revising the manuscript.

Major comments

Title

Accurately reflects the content of the manuscript

Abstract

The results found that hypoalbuminemia, inflammatory status (CRP > 10 mg/dL or IL-6 ≥ 10 28 pg/mL), and female gender was associated with increased, while age ≥ 85 yrs with decreased factors of seropositivity. So it would be more appropriate to change the last sentence as follows

Line 30-31

“Multivariable logistic regression revealed that hypoalbuminemia, inflammatory 30 status, and age ≥ 85 yrs were independent risk factors of seropositivity.” 

  Multivariable logistic regression revealed that hypoalbuminemia, inflammatory 30 status, and age ≥ 85 yrs were independently associated factors of seropositivity

Introduction

I think that it seems to be hard to understand what the 60+ notation means. So it would be more appropriate to change the words as follows

Line 53-54

The Turkish study (N = 941) with a low percentage of 53 older subjects (60+: 14.3%) showed a prevalence of 1.0%

The Turkish study (N = 941) with a low percentage of 53 older subjects (60 yrs ≤:  14.3%) showed a prevalence of 1.0%.

Materials and Methods

Line 126

“Chronic pain was defined as pain, that lasted more than 3 months [37]”.

The authors assessed the presence or absence of chronic pain in the participants.

Did the author not evaluate the body part where the pain is noted? Is visceral pain included?

If not, this may be another limitation of the study.

Results

Figer2

It is better to include 5 in the value of memory on the horizontal axis in the female group.

Line 222-223

It would be more appropriate to change the last sentence as follows

Multivariable logistic regression revealed that hypoalbuminemia, inflammatory status, and 222 age ≥ 85 yrs were independent risk factors of seropositivity

Multivariable logistic regression revealed that hypoalbuminemia, inflammatory status, and 222 age ≥ 85 yrs were independently associated factors of seropositivity

Discussion

Line 273-274

The authors described that anti-CCP antibodies can be detected in numerous disease entities other than RA. Prior studies showed the relationship between periodontal disease and anti-CCP antibody expression. It would have been better to study these relationships in this research. 

Author Response

The authors would like to thank the reviewers for their efforts to improve the paper.

Answers to the reviewer’s remarks are provided below.

  1. Abstract

・The results found that hypoalbuminemia, inflammatory status (CRP > 10 mg/dL or IL-6 ≥ 10 pg/mL), and female gender was associated with increased, while age ≥ 85 yrs with decreased factors of seropositivity. So it would be more appropriate to change the last sentence as follows

Line 30-31

“Multivariable logistic regression revealed that hypoalbuminemia, inflammatory 30 status, and age ≥ 85 yrs were independent risk factors of seropositivity.” 

→  Multivariable logistic regression revealed that hypoalbuminemia, inflammatory 30 status, and age ≥ 85 yrs were independently associated factors of seropositivity

Corrected as suggested.

  1. Introduction

・I think that it seems to be hard to understand what the 60+ notation means. So it would be more appropriate to change the words as follows

Line 53-54

The Turkish study (N = 941) with a low percentage of 53 older subjects (60+: 14.3%) showed a prevalence of 1.0%

→The Turkish study (N = 941) with a low percentage of 53 older subjects (60 yrs ≤:  14.3%) showed a prevalence of 1.0%.

Corrected as suggested.

  1. Materials and Methods

・Line 126

“Chronic pain was defined as pain, that lasted more than 3 months [37]”.

The authors assessed the presence or absence of chronic pain in the participants.

Did the author not evaluate the body part where the pain is noted? Is visceral pain included?

If not, this may be another limitation of the study.

Visceral pain was not included. The location of pain was analyzed.  We added: ‘The most frequent locations of pain in seropositive subjects were the lumbar region (28%), knees (22%)  and feet (22%).’

  1. Results

Figer2

・It is better to include 5 in the value of memory on the horizontal axis in the female group.

We have modified the figure according to the suggestion.

  1. Line 222-223

・It would be more appropriate to change the last sentence as follows

Multivariable logistic regression revealed that hypoalbuminemia, inflammatory status, and 222 age ≥ 85 yrs were independent risk factors of seropositivity

→Multivariable logistic regression revealed that hypoalbuminemia, inflammatory status, and 222 age ≥ 85 yrs were independently associated factors of seropositivity

 Corrected as suggested.

  1. Discussion

Line 273-274

・The authors described that anti-CCP antibodies can be detected in numerous disease entities other than RA. Prior studies showed the relationship between periodontal disease and anti-CCP antibody expression. It would have been better to study these relationships in this research. 

There were high rates of partial and complete edentulism in the PolSenior 1 population (45.7% and 47.1%, respectively). The data were previously published (Mehr K, Olszanecka-Glinianowicz M, Chudek J, Szybalska A, Mossakowska M, Zejda J, Wieczorowska-Tobis K, Grodzicki T, Piotrowski P. Dental status in the Polish senior population and its correlates-Results of the national survey PolSenior. Gerodontology. 2018; 35: 398-406.). We are sorry, but we did not assess the occurrence of periodontitis. We can’t discuss the problem of edentulism in our cohort.

We add that periodontitis related to the infection with Porphyromonas gingivalis is concern as the cause of induction of anti-CCP formation. We add:

“Moreover it is suggested, that the occurrence of periodontitis may predispose to the formation of anti-CCP antibodies [48]. However the periodontal status was not assessed, we cannot exclude that decreasing seropositivity is not related to the high rate of edentulism in the oldest [49]. “

Reviewer 2 Report

Novel study in Poland and other Central European countries and I have the following comments 

1.     Remove the subsections in the Abstract part.

2.     Avoid the use of abbreviation manner in the abstract part to be your paper clear for readers and easy to understand.

3.     The authors have to mention the p values in the results subsection of the abstract.

4.     Add to the conclusion subsection of the abstract in your own words your recommendation and required future work on anticitrullinated protein antibodies in older Poles.

5.     Many sentences kindly required to be rewritten to be clear for readers.

6.     Replace “To the best of our knowledge, there is no data concerning the prevalence of anti-CCP antibodies in Poland and other Central European countries. In this study, we analyze the prevalence and associates of anti-CPP antibodies occurrence in the Polish older population” with “To the best of our knowledge, there is no data concerning the prevalence of anti-CCP antibodies in Poland and other Central European countries. Therefore the current investigation aims to analyze the prevalence and associates of anti-CPP antibodies occurrence in the Polish older population”

7.     Kindly remove positive anti-CCP from the head of figure # 2 as it is mentioned in the figure title “Figure 2. Percentage of anti-CCP positive males and females in age-related subgroups.”.

8.     Improve the conclusion part and kindly avoid using in it the listing manner.

9.     The manuscript needs major grammar, typos, and editing corrections by a native speaker specialist in the biomedical sciences.

Author Response

The authors would like to thank the reviewers for their efforts to improve the paper.

Answers to the reviewer’s remarks are provided below.

  1. Remove the subsections in the Abstract part.

 Corrected as suggested.

  1. Avoid the use of abbreviation manner in the abstract part to be your paper clear for readers and easy to understand.

 Corrected as suggested.

  1. The authors have to mention the p values in the results subsection of the abstract.

 Corrected as suggested.

  1. Add to the conclusion subsection of the abstract in your own words your recommendation and required future work on anticitrullinated protein antibodies in older Poles.

The concussion was added.

  1. Many sentences kindly required to be rewritten to be clear for readers.

Replace “To the best of our knowledge, there is no data concerning the prevalence of anti-CCP antibodies in Poland and other Central European countries. In this study, we analyze the prevalence and associates of anti-CPP antibodies occurrence in the Polish older population” with “To the best of our knowledge, there is no data concerning the prevalence of anti-CCP antibodies in Poland and other Central European countries. Therefore the current investigation aims to analyze the prevalence and associates of anti-CPP antibodies occurrence in the Polish older population”

Corrected as suggested.

  1. Kindly remove positive anti-CCP from the head of figure # 2 as it is mentioned in the figure title “Figure 2. Percentage of anti-CCP positive males and females in age-related subgroups.”.

We have removed the title in the figure according to the suggestion.

  1. Improve the conclusion part and kindly avoid using in it the listing manner.

 We have modified the conclusions.

“The decreased frequency of anti-CCP seropositivity in the oldest old suggests shorter survival of the seropositive individuals with rheumatoid arthritis. It seems that low symptomatic RA remains frequently undiagnosed in older subjects.”

  1. The manuscript needs major grammar, typos, and editing corrections by a native speaker specialist in the biomedical sciences.

The paper was corrected by an English expert.

Reviewer 3 Report

Comments to authors

In this work, authors presented a study in which the prevalence of antibodies anti-cyclic citrullinated peptides (anti-CCP) was determined in old population from Poland. Alongside the determination of anti—CCP antibodies, several additional blood parameters, social data, and information regarding wealth status were recorded. The study was conducted on a sub-group consisting of roughly 1500 individuals from a larger study comprising roughly 5000 individuals.

Thanks to statistical interpretation authors found that three factors are linked to anti-CCP seropositivity, indeed female gender, increased levels of C-reactive proteins, increased hypoalbuminemia and increased interleukin-6 levels are risk factors that increase the risk of seropositivity.

Results were discussed in relation to rheumatoid arthritis.

This article is a sort of snapshot of the older Polish population. Many parameters have been recorded and the study is similar to other ones conducted in other countries. The novelty of this article is poor, however it reports data that can be useful for the country of origin.

I think the overall quality is acceptable, but it could be improved.

Here are listed some minor comments that could be addressed by authors:

1. line 38: Please change in “They are detected…”

2. lines 46-50: the presence of anti-CCP in several different diseases attests that this marker is not specific for rheumatoid arthritis. The lack of data regarding joint status, rheumatoid factor or other indicators is probably the weakest point of the study. Please comment it even in the introduction section, not only in the discussion.

3. lines 58-59 and 62-63: I think it should be written in another way such as, for example, “few parameters associated with” and “prevalence and associated parameters with anti-CCP”.

4. line 85: when first mentioned, it should be written 25-hydroxycholecalciferol and then its abbreviation.

5. lines 102-103: please change m2 in m2.

6. line 162: Nominal and ordinal data were compared with “2 test”. Please specify which test.

7. line 173: Please specify heart failure and then use the abbreviation HF.

8. lines 168-175: the comparison is clear only when looking at Table 1. Indeed in the text it is not specified that data of the sub-group were compared to that of the entire study group. Please mention it at the beginning of the paragraph otherwise it is difficult to understand (e.g. “There were more rural residents” with respect to the whole study group).

9. line 192-195: These data seem to be in contrast with previous studies mentioned in the introduction. This was mentioned by authors in the discussion. However, this higher prevalence of anti-CCP observed in Polish population by authors is commented only with respect to premature mortality even though the survival curve was not affected. Please comment.

10. line 221: alcohol consumption seems to be correlated to a decreased risk of seropositivity to ant-CCP. Please comment this finding in the discussion.

11. Lines 286-287: I do not agree with this conclusion. Please better explain or modify this statement given that no significant correlation was found with survival rate (as suggested in  lines 227-229 and in lines 244-248).

12. there are too many words which need to be corrected due to the presence of an hyphenation (e.g. lines 67, 69, 76, 87, 91, 93, ….). Please check the entire manuscript

Author Response

The authors would like to thank the reviewers for their efforts to improve the paper.

Answers to the reviewer’s remarks are provided below.

  1. line 38: Please change in “They are detected…”

Corrected as suggested.

  1. lines 46-50: the presence of anti-CCP in several different diseases attests that this marker is not specific to rheumatoid arthritis. The lack of data regarding joint status, rheumatoid factor or other indicators is probably the weakest point of the study. Please comment it even in the introduction section, not only in the discussion.

We have added that: ‘’ Anti-CCP antibodies cannot be considered as pathognomonic for the RA, as they are also detected in other disease entities such as palindromic rheumatism, psoriatic arthritis, systemic lupus erythematosus, systemic sclerosis, Sjogren’s syndrome, polymyositis, dermatomyositis, ulcerative colitis, HCV seropositive patients with arthralgia and also in some infectious diseases  [10 - 22].”

  1. lines 58-59 and 62-63: I think it should be written in another way such as, for example, “few parameters associated with” and “prevalence and associated parameters with anti-CCP”.

Corrected as suggested.

  1. line 85: when first mentioned, it should be written 25-hydroxycholecalciferol and then its abbreviation.

Corrected as suggested.

  1. lines 102-103: please change m2 in m2.

Corrected as suggested.

  1. line 162: Nominal and ordinal data were compared with “2 test”. Please specify which test.

Corrected as suggested.

  1. line 173: Please specify heart failure and then use the abbreviation HF.

We assessed hospitalizations for congestive heart failure. It was corrected across the paper.

  1. lines 168-175: the comparison is clear only when looking at Table 1. Indeed in the text it is not specified that data of the sub-group were compared to that of the entire study group. Please mention it at the beginning of the paragraph otherwise it is difficult to understand (e.g. “There were more rural residents” with respect to the whole study group).

We have corrected the text according to the suggestion.

  1. line 192-195: These data seem to be in contrast with previous studies mentioned in the introduction. This was mentioned by authors in the discussion. However, this higher prevalence of anti-CCP observed in Polish population by authors is commented only with respect to premature mortality even though the survival curve was not affected. Please comment.

We have commented that it may be caused by a small subset of seropositive subjects: ‘However, our 5-year survival analysis failed to support the shorter survival of seropositive subjects, which may be caused by a small subset of anti-CPP positive, declining the power of statistical analysis.’

  1. line 221: alcohol consumption seems to be correlated to a decreased risk of seropositivity to ant-CCP. Please comment on this finding in the discussion.

The multiple regression analysis did not support the independent association between alcohol consumption and seropositivity. Therefore there is much to discuss.

  1. Lines 286-287: I do not agree with this conclusion. Please better explain or modify this statement given that no significant correlation was found with survival rate (as suggested in lines 227-229 and in lines 244-248).

We have corrected as follows:

Due to the small sample size, we did not analyze the influence of age and the presence of rheumatoid arthritis symptoms on survival.

And we changed the conclusion: The decreased frequency of anti-CCP seropositivity in the oldest old suggests shorter survival of the seropositive individuals who developed rheumatoid arthritis.

  1. there are too many words which need to be corrected due to the presence of an hyphenation (e.g. lines 67, 69, 76, 87, 91, 93, ….). Please check the entire manuscript

Corrected as suggested.

Round 2

Reviewer 1 Report

The revised text seems to adequately address the questions posed by the reviewers. It seems to be appropriate for publication in “ijerph”.

Reviewer 2 Report

The authors conducted all the required corrections, thank you